# Iridium-catalyzed direct asymmetric reductive amination utilizing primary alkyl amines as the N-sources

Zitong Wu[1,2,4], Wenji Wang [1,4], Haodong Guo[1], Guorui Gao[3], Haizhou Huang [1] & Mingxin Chang [1,2✉]

Direct asymmetric reductive amination is one of the most efficient methods for the construction of chiral amines, in which the scope of the applicable amine coupling partners remains a significant challenge. In this study we describe primary alkyl amines effectively serve as the N-sources in direct asymmetric reductive amination catalyzed by the iridium precursor and sterically tunable chiral phosphoramidite ligands. The density functional theory studies of the reaction mechanism imply the alkyl amine substrates serve as a ligand of iridium strengthened by a (N)H-O(P) hydrogen-bonding attraction, and the hydride addition occurs via an outer-sphere transition state, in which the Cl-H H-bonding plays an important role. Through this concise procedure, cinacalcet, tecalcet, fendiline and many other related chiral amines have been synthesized in one single step with high yields and excellent enantioselectivity.

[1] College of Chemistry & Pharmacy, Northwest A&F University, Yangling, Shaanxi 712100, China. [2] College of Plant Protection, Shaanxi Research Center of Biopesticide Engineering & Technology, Northwest A&F University, Yangling, Shaanxi 712100, China. [3] College of Chemistry, Chemical Engineering and Materials Science, Collaborative Innovation Center of Functionalized Probes for Chemical Imaging in Universities of Shandong, Shandong Normal University, 88 Wenhuadong Road, Jinan 250014, China. [4]These authors contributed equally: Zitong Wu, Wenji Wang. ✉email: mxchang@nwsuaf.edu.cn

Enantiomerically enriched *N*-alkyl amines are common structural motifs for active pharmaceutical ingredients (Fig. 1a). The increasing demand has driven the development of novel and efficient methods for their synthesis[1–3], including two practical and highly efficient routes, asymmetric hydrogenation[4–9] and reductive amination[10–15] (Fig. 1b). Direct asymmetric reductive amination (DARA) is one of the most efficient approaches for the construction of chiral amines by enabling the coupling of ketones with amines in a single step rather than taking a circuitous route through imine/enamine preparation, reduction, and/or the following *N*-deprotection. For the transition-metal-catalyzed DARA, as some progress has been made, the applicable scope of the amine coupling partners are still very limited. Apart from the intramolecular DARAs[16–23], most reported N-sources of DARAs generally fall into three categories: inorganic ammonium salts[24–29]/ammonia[30] for the synthesis of primary amines; aryl amines[31–35], the popularly applied N-sources, albeit often times the *N*-Ar group in the product is not desired and needs to be removed, which lead to secondary chiral amine products; and one example of secondary amine sources for the construction of tertiary chiral amine products[36]. Besides, there are some other sporadically reported special amines, including benzyl amines[37,38], diphenylmethylamine[39–41], hydrazine[42], and hydrazides[43,44].

Previously there was only one example that utilized an alkyl amine, MeNH$_2$, as the N-coupling partners in transition-metal-catalyzed DARAs[45], and some instances in which special alkyl amines, such as benzyl amines[38,39] diphenylmethylamine[39–41], and allylamine[46], were used. Even the literature for the asymmetric hydrogenation of corresponding *N*-alkyl imines to directly form the *N*-alkyl amines are extraordinary scarce[47–52], of which in two cases the *N*-alkyl amines were in situ transformed into *N*-SiH$_2$Ph and *N*-Boc products in order to decrease the inhibitory effects on the catalysts[49–51]. One of the main reasons for this limitation in DARA research is that the alkyl groups could not coordinate with or interact with the functioning catalytic species via secondary interactions, for example, hydrogen-bonding, π effects, and electrostatic interactions.

The inhibitory effects of nitrogen-containing species, especially the electron-rich *N*-alkyl amines and imines in this case, on the catalyst is also a major factor. Another difficulty for lagging the effective utility of primary alkyl amines in DARAs is the direct products are secondary amines, which could serve as new coupling partners to continue to react with ketone substrates and form tertiary amines[36]. We postulate that the application of readily steric tuning chiral ligands, which are capable of efficiently confining the position of the alkyl groups during the catalytic process, along with the accelerations from the additives, may tackle this problem.

Herein we report a highly efficient DARA of various ketones with primary alkyl amines for the synthesis of corresponding chiral secondary *N*-alkyl amines catalyzed by 0.05 mol% iridium-phosphoramidite complex (Fig. 1c). The evolved phosphoramidite chiral ligands with bulky 3,3′-1-naphthyl substituents successfully manage the enantioselective process of the reaction with the lack of secondary interactions between the alkyl groups and the catalytic complex. Moreover, computational studies have been carried out to provide useful information about the reaction mechanism and reveal an outer-sphere hydride addition pathway, in which two H-bonding attractions, one between (P)O of the phosphoramidite chiral ligand and (N)H and the other between the chlorine on iridium and the imine substrate, play important roles.

## Result and discussion
### Steric tuning of the chiral monodentate phosphoramidite ligands.
The present study was initiated by the direct reductive amination of acetophenone **1** and 3-phenylpropylamine **2**, to mimic an alkyl amine, which was catalyzed by the catalyst in situ generated from [Ir(cod)Cl]$_2$ and H$_8$-BINOL-based chiral ligand **L1** (Table 1). The monodentate phosphoramidite ligands have found success in numerous catalytic research areas because of their high degree of tunability on both electronic and steric properties, and ease of preparation at low cost[53,54]. From the brief solvent screening we can see, although multiple solvents led to excellent yields for the reaction, only protic methanol and trifluoroethanol displayed enantioselectivity moderately (Table 1, entries 1–6).

To sufficiently exploit the readily fine-tuning feature of this kind of ligands, we deliberately modified the H$_8$-BINOL backbone by introducing methyl, phenyl, and 1-naphthyl groups at the 3,3′-positions and synthesized ligands **L2**, **L3**, and **L4**. Compared with the results from **L1**, the reaction yield and enantioselectivity of **L2** decreased by 17% and 11% respectively, indicating that the methyl groups at the 3,3′-positions compromised the proper coordination of the substrate to the transition-metal complex. While the substituents were enlarged into phenyl groups, the ee value was improved significantly to 69% (Table 1, entry 8). To further enlarge the size of imbedded segments to 1-naphthyl groups (**L4**), the enantioselectivity was elevated to 86% (Table 1, entry 9). The improved stereoselectivity indicates the spacious 1-naphthyl groups on **L4** could effectively confine the imine substrate for better steric differentiation (Fig. 2)[55]

Additives in DARA reactions have been proven to be perfectly competent in improving both reaction reactivity and selectivity significantly[31–44,56–58]. Titanium isopropoxide is well known to be able to facilitate the formation of the imine intermediates[59]. Some other common additives, iodine, Brønsted/Lewis acids/bases, and non-coordinate anion were also evaluated (Table 1, entries 10–14). Among them, Lewis acid FeCl$_3$ effectively increased the reaction yield and stereoselectivity (Table 1, entry 14). Further solvent screening revealed the combination of trifluoroethanol and methyl acetate provided a satisfying ee value at 97% with an excellent reaction yield. We speculated that it was Brønsted acid HCl generated from the hydrolysis of Lewis acid FeCl$_3$ in the reaction system that successfully promoted the stereoselectivity. This was proved by the result from the mere addition of 30 mol% hydrochloride salt of **2** (Table 1, entry 17). When the catalyst loading was reduced to 0.02 mol%, the reaction still proceeded smoothly without significant erosion in reactivity and selectivity with 88% yield and 95% ee (Table 1, entry 19).

**Substrate scope.** Based on the optimal conditions for our iridium-catalyzed DARA, firstly the scope of aromatic ketones was probed with 3-phenylpropylamine **2** as a representative amine partner applying 0.05 mol% the in situ generated iridium catalyst (Fig. 3a). In cases the enantiomers of the products were unable to be separated by available chiral HPLC columns, benzylamine was used instead as the coupling partner of the ketones. At first, we examined the effects of the electronic property and steric hindrance on the reactions for the substituted arylethanones. It appeared the common substituents including small-sized groups at the ortho-position had no significant influence on the reaction yields or stereoselectivity (products **3–21**), regardless of their electron-withdrawing or donating properties. It is worth mentioning that for some steric-hindered substrates, the corresponding products were still obtained in excellent ee values (products **22–23**, **26–29**). To achieve better results for space-demanding products **27–29**, modified chiral ligand **L5** (Fig. 3) was applied and displayed higher stereoselectivity with the opposite spatial configuration than **L4**, further demonstrating the

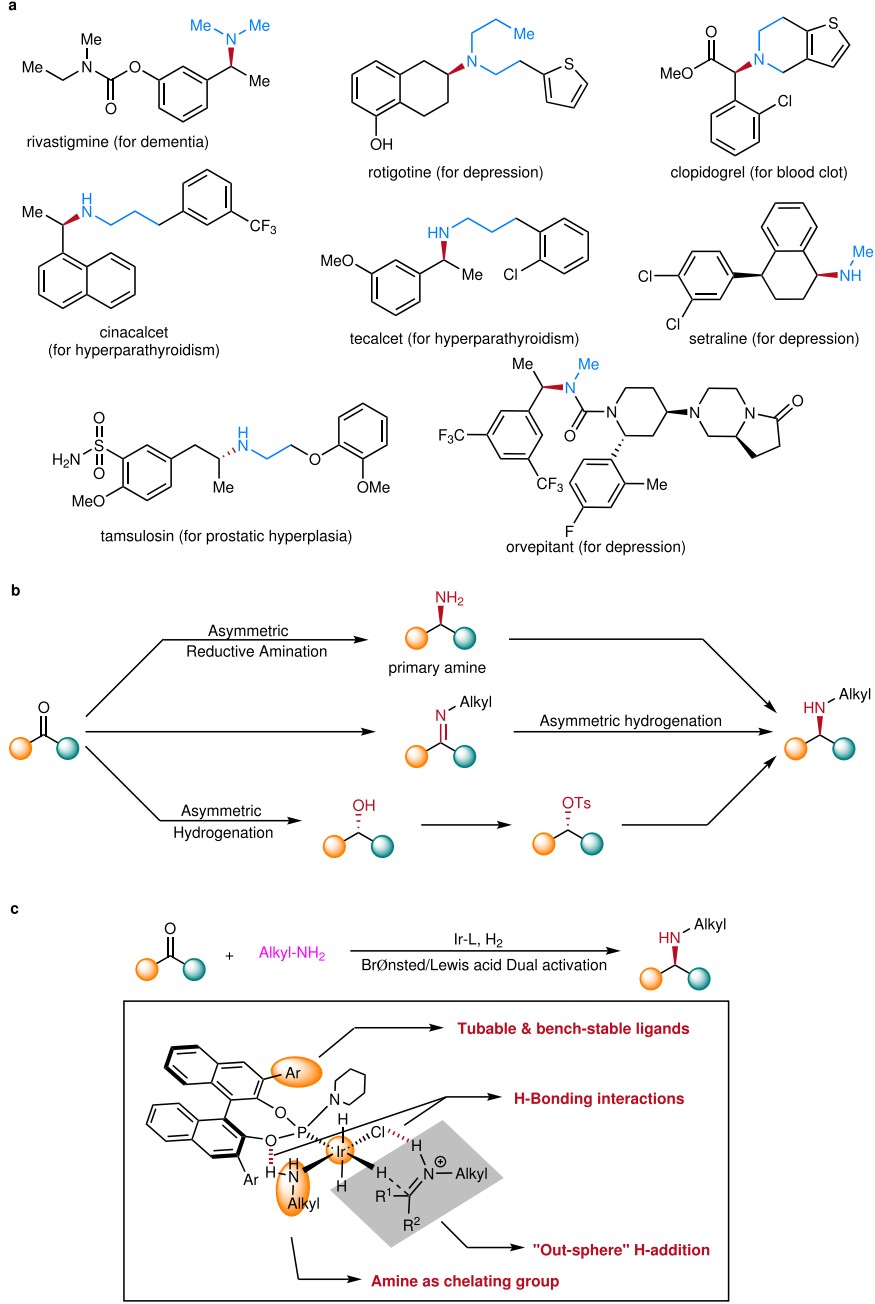

**Fig. 1 Relevance of chiral *N*-alkyl amines and efficient enantioselective synthetic strategies. a** Representing *N*-alkyl pharmaceuticals. **b** Practical methods for the synthesis *N*-alkyl amines. **c** This work: primary alkyl amines as N-sources for DARA.

versatility of this class of ligands. To our delight, various functional groups, including nitro (-NO$_2$), benzyloxy (-OBn), thioether (-SMe), tert-butyl carbamate (-NHBoc), borate ester, and ester, were all well-tolerated during this transformation, in which the corresponding products **30–35** were obtained in good yields and excellent ees. Heteroaromatic ketone 2-acetylthiophene could be employed in this reaction to provide **36** in a high yield and good enantioselectivity.

Next, the scope of primary aliphatic amines was explored with acetophenone or 1′-acetonaphthone as the representative ketone partners (Fig. 3b). It is important to note that benzylamine was a valuable coupling partner for the DARA with **1** leading to the corresponding N-Bn product, which could be easily deprotected under mild conditions to afford α-phenylethylamine[60,61], a versatile building-block and common resolution reagent in

organic synthesis. The aromatic amine aniline as a weaker nucleophile was also a suitable N-source for the reductive coupling reaction with acetophenone to provide excellent results. The DARAs of **1** and pure aliphatic amine sources bearing linear or cyclic alkyl groups also proceeded smoothly to afford products **40–46** with excellent enantioselectivity and yields. Importantly, N-sources with heteroaromatic segments, such as furan, thiophene, indole, and pyridine, reductively coupled with acetophenone successfully to provide products **47–50**. Besides, the catalytic system effectively tolerated NHBoc, olefin, and ether functional groups (products **51–53**).

After having successfully conducted the DARA reactions of aromatic ketones with amines, we were interested in the applicability of this iridium catalysis in alkyl ketones for the synthesis of chiral aliphatic amines, which are difficult to

**Table 1 Initial reaction development[a].**

| Entry | ligand | Solvent | Additive[b] | Yield (%) | ee (%) |
|---|---|---|---|---|---|
| 1 | L1 | CH$_2$Cl$_2$ | - | 14 | 0 |
| 2 | L1 | PhMe | - | 79 | 0 |
| 3 | L1 | EtOAc | - | 97 | - |
| 4 | L1 | MeOH | - | 98 | 22 (R) |
| 5 | L1 | THF | - | 97 | 2 |
| 6 | L1 | TFE | - | 87 | 29 (S) |
| 7 | L2 | TFE | - | 70 | 18 (S) |
| 8 | L3 | TFE | - | 94 | 69 (S) |
| 9 | L4 | TFE | - | 80 | 86 (S) |
| 10 | L4 | TFE | I$_2$ | 90 | 81 (S) |
| 11 | L4 | TFE | TsOH | 95 | 72 (S) |
| 12 | L4 | TFE | Et$_3$N | 95 | 71 (S) |
| 13 | L4 | TFE | NaBF$_4$ | 92 | 56 (S) |
| 14 | L4 | TFE | FeCl$_3$ | 88 | 89 (S) |
| 15 | L4 | TFE | CsCl | 78 | 86 (S) |
| 16 | L4 | TFE/MeOAc | FeCl$_3$ | 96 | 97 (S) |
| 17[c] | L4 | TFE/MeOAc | HCl | 96 | 95 (S) |
| 18[d] | L4 | TFE/MeOAc | FeCl$_3$ | 99 | 98 (S) |
| 19[e] | L4 | TFE/MeOAc | FeCl$_3$ | 88 | 95 (S) |

*TFE* 2,2,2- trifluoroethanol, *TsOH* p-toluenesulfonic acid.
[a]Reaction conditions: [Ir]/**L**/**1**/**2** = 1:2.1:100:100; **1** 0.1 mmol, solvent 2 mL, r.t., 24 h. Yields and enantiomeric excesses were determined by chiral HPLC after the products were converted to the corresponding acetamides.
[b]The amount of the added additives was 10 mol%.
[c]The 3-phenylpropylamine hydrochloride salt (30 mol%) was used instead of FeCl$_3$.
[d]The catalyst loading was 0.05 mol%. The reaction temperature was 40 °C. FeCl$_3$ was 30 mol%.
[e]The catalyst loading was 0.02 mol%. The reaction temperature was 60 °C. FeCl$_3$ was 30 mol%.

**Fig. 2 Structural tuning of the phosphoramidite ligands. a** Structures of developed ligands. **b** Ligand–substrate interactions.

synthesize via asymmetric catalytic methods, due to the lack of aforementioned secondary interactions between substrates and the catalytic complex[62,63]. To date, the successful DARA examples of alkyl–alkyl ketones are very scarce, in which some special amine partners, anilines, diphenylmethanamine, and benzhydrazide, were utilized to gain additional anchors with the catalysts[31–35,40]. Accomplishment in the reductive coupling of aromatic ketones prompted us to validate this protocol for more challenging alkyl–alkyl ketone substrates. Fortunately, the iridium-**L4** catalytic system is found to have well functioned in

this substrate category (Fig. 4a). Accordingly, various aliphatic ketones with the linear, branch, and cyclic segments prosperously reacted with alkyl amines to effectively furnish the chiral aliphatic amines **54–61** in excellent yields and enantioselectivities. Importantly, this procedure was also very effective towards ketones and amines with pure alkyl components to afford the all-alkyl amines **59–61**. **L6** was utilized in the synthesis of products **60** and **61** for higher ee values, which again displayed the advantage of the readily fine-tuning property of this type of ligand.

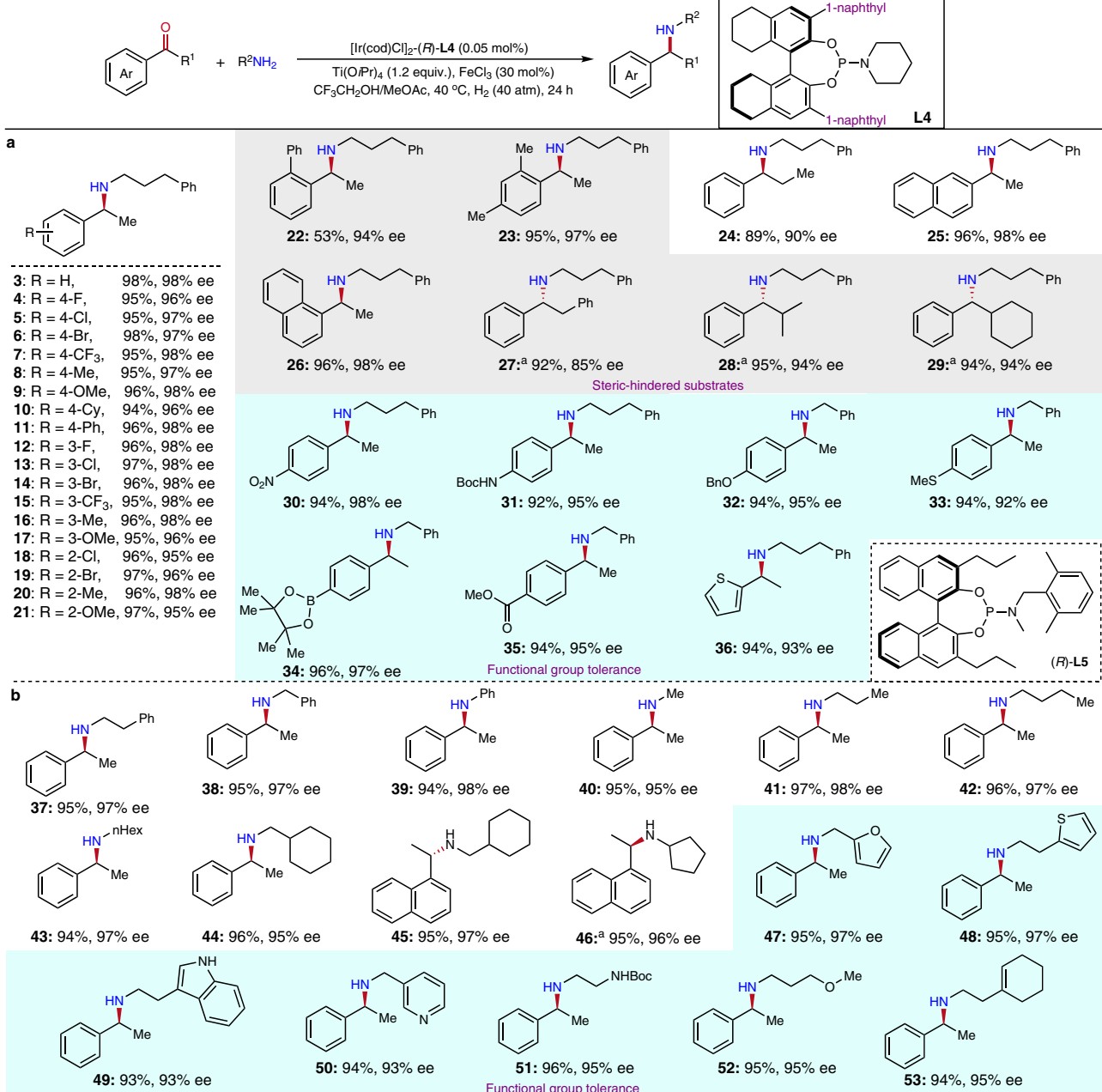

**Fig. 3 The DARA reactions of aromatic ketone with alkyl amines. a** Aromatic ketone scope. **b** Amine scope. Conditions: [Ir]-**L4** 0.05 mol%; ketone 0.3 mmol, amine 0.95 equiv. (MeNH₂ and nPrNH₂ were 1.5 equiv.), Ti(OiPr)₄ 1.2 equiv., FeCl₃ 30 mol%, solvent (CF₃CH₂OH/MeOAc 1:1) 1.2 mL, 40 °C, 24 h; Yields were isolated yields; Enantiomeric excesses were determined by chiral HPLC after the products were converted to the corresponding acetamides. [a]Reaction conditions: [Ir]-**L5** 0.1 mol%, Ti(OiPr)₄ 1.2 equiv., 1,8-diazabicyclo[5.4.0]undec-7-ene 10 mol%, Et₃N·HCl 30 mol%, MeOAc 2 mL, 60 atm H₂, 60 °C, 24 h.

**Practical applications**. To showcase the practical utility, a 10-gram-scale reaction was carried out (Fig. 4b). Catalyzed by 0.05 mol% of the Ir-**L4** complex, 10 grams of acetophenone efficiently reductively coupled with 8.5 grams of benzylamine to afford 16.7 grams of N-benzyl-1-phenylethan-1-amine **38** in 94% yield and 95% ee, which were similar to the results from the small-scale reaction. To further demonstrate potential, this iridium catalysis was applied in the synthesis of a collection of life-science important pharmaceuticals and key intermediates. Utilizing 3-(3-(trifluoromethyl)phenyl) propan-1-amine as the amine source, the calcimimetic agent cinacalcet could be synthesized at gram-scale via this protocol in 96% yield and 94% ee. Similarly, tecalcet and fendiline were prepared in excellent ee's

and high yields. With methyl amine as the coupling partner, the key intermediates of rivastigmine and orvepitant, **62** and **66**, were effectively constructed.

**Mechanistic studies**. To gain insights into the complete reaction features of this iridium catalysis, we performed DFT calculations using Gaussian 09 program[64] at the B3LYP-D3 level of theory with the 6–311 G(d,p) basis set for C, P, N, O, F, Cl, H and LANL2DZ for Ir. Since we utilized a mixed solvent in the experiment, we separately carried out the calculation in both solvents and found the energies in CF₃CH₂OH (see Supplementary Data 1) are commonly lower than in EtOAc

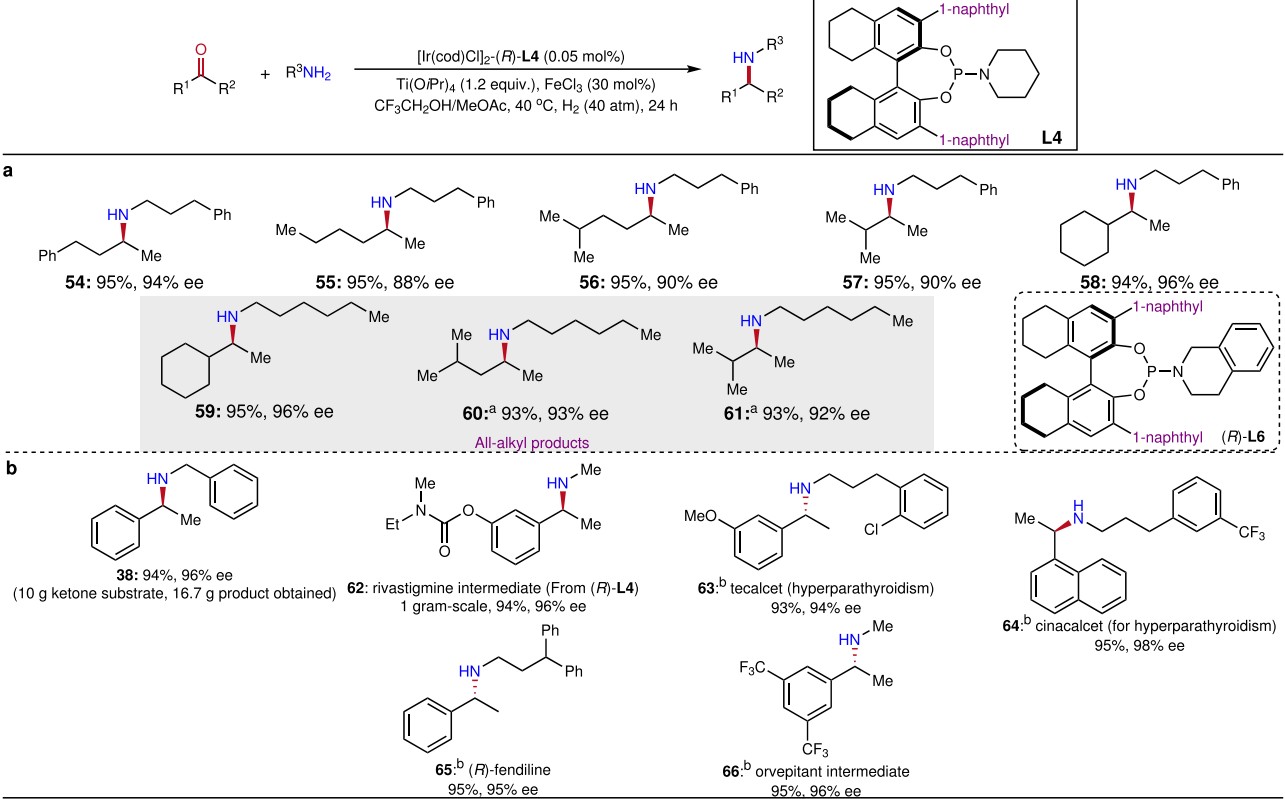

**Fig. 4 The DARA reactions of aliphatic ketone with alkyl amines and applications. a** Aliphatic ketone scope. **b** Applications in drugs and key intermediates synthesis. Conditions: [Ir]-**L4** 0.05 mol%; ketone 0.3 mmol, amine 0.95 equiv. (MeNH₂ was 1.5 equiv.), Ti(O*i*Pr)₄ 1.2 equiv., FeCl₃ 30 mol%, solvent (CF₃CH₂OH/MeOAc 1:1) 1.2 mL, 40 °C, 24 h; Yields were isolated yields; Enantiomeric excesses were determined by chiral HPLC or GC after the products were converted to the corresponding acetamides (for product **59**, it was converted to benzamide). For substrates in Fig. 4a, FeCl₃ 0 mol%, amine 1.1 equiv., solvent (CF₃CH₂OH/MeOAc 1:5) 1.2 mL. ᵃ(*R*)-**L6** was used instead of **L4**. ᵇ(*S*)-**L4** was used instead of **L4**.

(see Supplementary Data 2). Based on the calculation results and literatures on iridium catalysis[65–68], we proposed a possible reaction pathway (Fig. 5a) and summarized the energy profiles (Fig. 5b). L3, acetophenone **1**, and methyl amine were employed to simplify the calculation process. Initially methyl amine coordinates to Ir and breaks the dimer **I**[69], in situ generated from [Ir(cod)Cl]₂ and **L3**, to form complex **II**, which is substantially stabilized by an H-bonding attraction between the (N)-H of methyl amine and (P)-O of the chiral ligand **L3** (Fig. 5b, **TS3-S**) with the H–O distance at 2.0 Å. This step is rather facile and exergonic by 12.8 kcalmol⁻¹, followed by the oxidative addition, in which H₂ is activated and adds to Ir(I) via transition state **TS1** to afford Ir(III) (intermediate **IV**). Then the incoming second H₂ molecule prefers to coordinate to the *cis*-position of P than the *trans*-position (see competing intermediates **IV′** and **IV″** in Fig. 5) to form complex **V**. And with the help of the imine substrate, it heterocleaves and the resulting hydride adds to Ir to form a tight ion pair complex[67] **VI** through transition state **TS2** with an activation barrier of 9.9 kcalmol⁻¹.

The origins of enantioselectivity occur in the hydride addition step. From the optimized transition state **TS3-S** we can see that the distance between the chloride ion on Ir and the proton on imine is 2.53 Å, indicating there is an H-bonding interaction (Fig. 5b). The hydride adds from the *Si*-face of the imine substrate (**TS3-S**) is thermodynamically favored by 1.6 kcalmol⁻¹ than from the *Re*-face (**TS3-R**), to afford the amine product and be back to complex **IV** with the release of 26.4 kcalmol⁻¹ of Gibbs free energy. In the competing transition state (**TS3-R′**), there exists a similar H-bonding interaction between one of the hydride ions on Ir and the proton on imine (H-H 2.22 Å, Fig. 5b). During

the catalytic cycle, the imine substrate does not coordinate directly with the iridium metal center, that is, the hydride addition is an outer-sphere process, in which the bulky aromatic groups at the 3,3′-positions of the chiral BINOL-based phosphoramidite ligands **L3** and **L4** contribute by forcing the imine substrate to the outer sphere of the catalytic complex.

We also calculated the Gibbs energy for the "inner-sphere" H-addition pathways, in which the oxidation state of iridium changes from +1 to +3 (see Supplementary Data 3). The two different pathways share early intermediates **II**, **III**, and **IV**, and transition state **TS1**. Compared with the single hydride addition transition state **TS3** of the "outer-sphere" pathway, there are two H-addition transition states for the "inner-sphere" one, **IS-TS2** and **IS-TS3**, through which two hydrides add to C and N of the imine substrate subsequently. The Gibbs free energy for the second transition state, **IS-TS3**, is much higher than that of **TS3**, thus enabling the "outer-sphere" alternative to occur more likely.

In summary, we have successfully applied primary alkyl amines as the N-sources in direct catalytic asymmetric reductive aminations of a broad range of ketones. A notable feature of the applied chiral phosphoramidite ligands is their tunability for accommodating our specific substrates. Using as low as 0.02 mol % catalyst, the developed procedure is effective towards both aromatic and aliphatic ketones, offering various chiral secondary alkyl amine in excellent enantioselectivity and high yields. A collection of pharmaceuticals and important intermediates were facially synthesized in one single step. The 10-gram-scale experiment further demonstrates the practical utility of this methodology. The DFT studies reveal the amine substrate serves as a ligand of the transition-metal center and the hydride addition

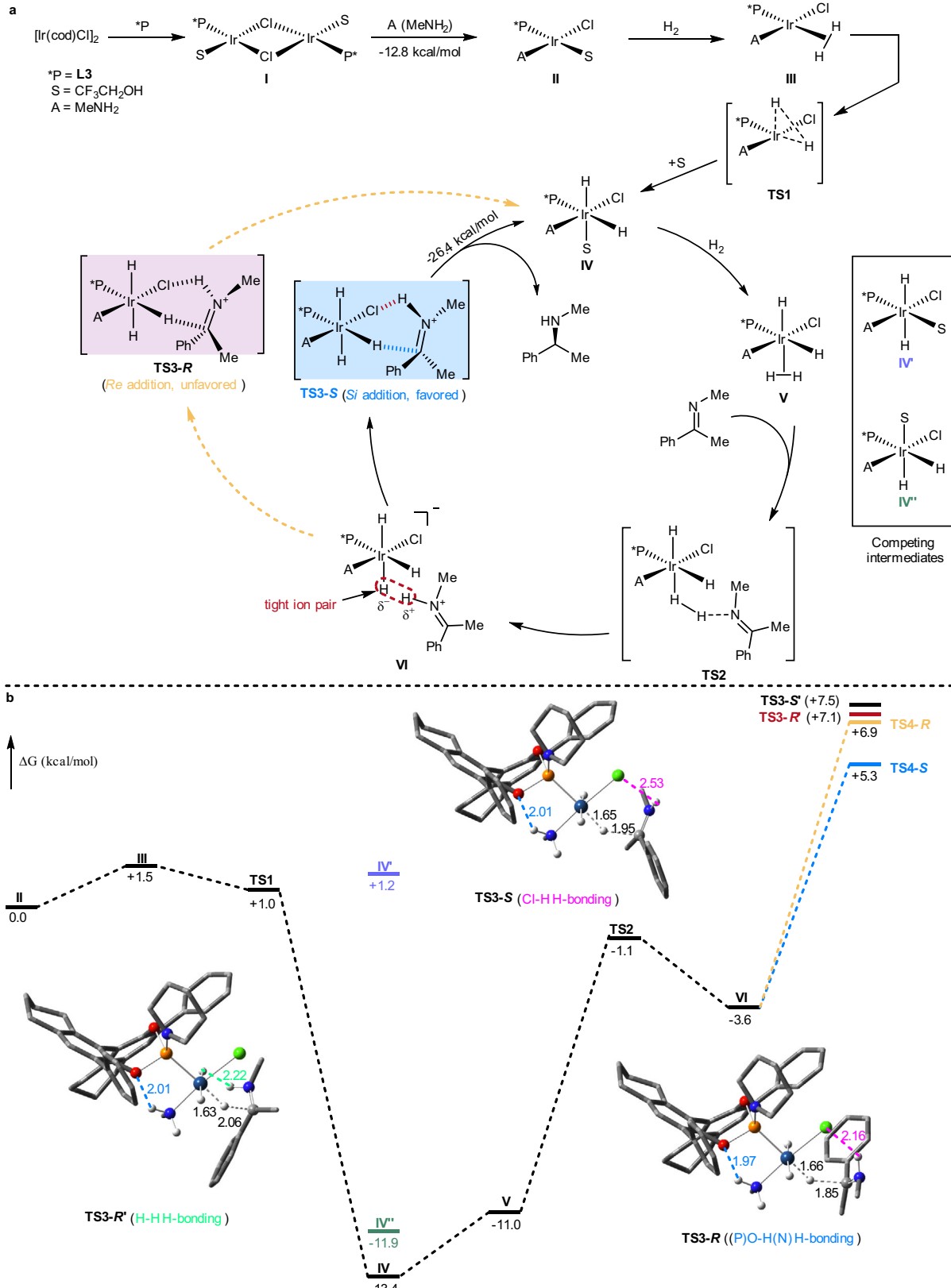

**Fig. 5 Proposed reaction pathways and energy profiles. a** Catalytic cycle. **b** Gibbs energy profiles.

occurs through an outer-sphere transition state, in which there are two H-bonding attractions, one between the (N)-H of amine substrate and (P)-O of the chiral ligand and the other between the chloride ion on Ir and the proton on the imine intermediate. Our

protocol greatly broadens the scope of the N-coupling partners for current asymmetric reductive amination research, opening the door for the direct and efficient synthesis of related chiral secondary amines.

## Methods

**General procedure for direct reductive amination**. In a nitrogen-filled glovebox, [Ir(cod)Cl]$_2$ (2.0 mg, 3 mmol) and **L4** (3.3 mg, 12.6 mmol) were dissolved in anhydrous trifluoroethanol (2 mL) in a 10 mL vial equipped with a stir bar. The above solution was stirred at room temperature for 20 min to in situ generate the Ir-**L4** complex. To a 5 mL vial equipped with a stir bar were added ketone (0.3 mmol) and amine (0.29 mmol, 0.95 equiv.) substrates, followed by the addition of anhydrous trifluoroethanol (0.5 mL), Ti(O*i*Pr)$_4$ (0.36 mmol, 1.2 equiv.), FeCl$_3$ (0.09 mmol, 30 mol%), and the solution of the Ir-**L4** complex (50 μL, 0.05 mol%). The total amount of solvent was made to 1.2 mL (CF$_3$CH$_2$OH/MeOAc = 1:1). The resulting vial was transferred to an autoclave, which was purged with H$_2$ 3 times and then charged with H$_2$ (40 atm), and stirred at 40 ºC for 24 h. After the reaction was complete, the hydrogen gas was released slowly and the reaction solution was concentrated to give the crude products, which were purified by column chromatography (silica gel, petroleum ether/EtOAc from 10/1 to 2/1 with 0.5% Et$_3$N) to afford the final product.

## Data availability

The authors declare that the data supporting the conclusions of this study are available within the article and its Supplementary Information file.

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

## Acknowledgements
Financial support from the National Natural Science Foundation of China (21772155, M.C.) and the Scientific Fund of Northwest A&F is gratefully acknowledged. We greatly acknowledge the HPC of Northwest A&F University for the DFT calculations carried out in this work.

## Author contributions
Z.W. and H.G. established the reaction conditions. W.W. performed the DFT calculations. Z.W., G.G., and H.H. expanded the substrate scope. M.C. conceived and supervised the project and wrote the manuscript. All the authors discussed the results and commented on the manuscript.

## Competing interests
The authors declare no competing interests.
