## [Peer Review File · Nature Communications]

REVIEWER COMMENTS

Reviewer #1 (Remarks to the Author):

In this manuscript, the authors describe a Ir-system for the direct asymmetric reductive amination (ARA) of various ketones with alkyl amines to the corresponding enantiomeric pure alkylamines. This is a significant advance in this field, as alkyl amines are very challenging substrates for high ee in the metal catalyzed ARA. The provided substrate scope as well as the enantioselectivities and yields are impressive. Especially the authors could also obtain remarkable high ee in the ARA of aliphatic ketones with aliphatic primary amines, a notorious difficult transformation in the ARA. This is a significant step in this field and could be very useful to be adapted in organic synthesis. Also, the reaction works already at relatively low Ir-loadings. The isolated material is well characterized in the supporting information. Therefore, I can strongly recommend that this results are published in nature communications, after addressing some points to further improve the manuscripts and make some points more clear to the reader.

The role of the stoichiometric additive $[\text{Ti}(\text{OiPr})_4]$ is not discussed at all or at least described, why it was added in all reactions. The authors should add some description, why this was added (most likely to force imine formation by water scavenging) and what happens without it.

On the role of the FeCl_3/HCl co-additives. The authors assume, that it's due to the bronsted acid behavior of the HCl or generated HCl from FeCl_3 which leads to the improved ee. But why is the ee not really increased, when adding the bronsted acid TsOH (entry 11 in table 1)? One could also assume, that there can be reactions with the $[\text{Ti}(\text{OiPr})_4]$ additive, as this is prone to hydrolysis. As the beneficial effect to improve the ee was only shown with the Cl-containing additives, may it is also the role of the Cl as a crucial ligand on the Ir to improve the ee? One can speculate, that by adding a Cl-source, the Ir-Cl species stay predominant, as the authors show in scheme 2, the Cl-ligand is crucial in the mechanism. It is also known from other ARA reports, that the halide ligand on the metal is of importance (see ref31). To validate this (or to rule this out), an experiment just adding a Cl-source without acidic properties as LiCl or CsCl would be of interest to see, if only adding Cl improves the ee, which could also be explained by the proposed mechanism. Or add HBr.

DFT calculations/Mechanistic studies: In the calculations (see supplementary information 4), it is described, that "Single point energy calculations were performed on optimized geometries in EthylEthanoate solvent using the IEFPCM model at the B3LYP-D3 level of theory." This is a very unpolar solvent considered for the calculations. Especially, as significant ee were only obtained when using polar and protic solvents as MeOH or TFE. TFE is a rather polar and protic solvent and as crucial intermediates in the proposed mechanism are charged and H-interactions are involved, one

can expect, that the energies would also be affected, if a more polar-protic solvent environment is considered in the DFT calculations (e.g. the proposed ion pairs should be less stable in a very polar-protic environment than in a unipolar solvent). The authors should at least discuss, why they assume, that this rather unipolar solvent environment in the DFT calculations will provide accurate energies for the proposed mechanism. But I would recommend recalculating using a more realistic solvent model.

In Scheme 2 and the according discussion, it is nowhere provided, what is the solvent coordinated in the species I, II and III. This should be provided. Species IV is a neutral Ir(III) dihydride. But after the H₂ activation, species VI and the following transition state it should be an anionic Ir(III)-trihydride. All relevant charges should be provided in the scheme.

There is a general inconsistency between the structural tuning in figure 1 and the proposed calculated mechanism in scheme 2: In figure 1, the structural tuning is described on square-planar tetrahedral Ir(I) (?) complexes, but in the proposed mechanism, all relevant Ir-species responsible for the enantioselectivity are octahedral Ir(III) complexes. In this context, was any pathway calculated or considered, where a H₂ activation via a Ir(I)-Ir(III) mechanism is involved (including a reductive elimination step)? Also, the pictures of provided calculated transition states TS4 should be improved, to provide the reader valuable information. In TS4-S the MeNH₂ ligand looks just like a NH₃. Where is the Methyl-group? To add some drawing (dashed lines) for the relevant bonding situation incl. bond length in TS4-R would be a useful information. A picture of the structure with the relevant bond length in VI would also be of interest. Is the positive charge still on the H, as drawn in the schematic picture? Seems rather unusual for a structure which is not a transition state.

Therefore, I would recommend, to revise the calculations according to the above given points and also provide (e.g. in the supporting information) further insights and alternative pathways considered. But this would result in a lot of additional work, may better address in a follow up paper focusing on the mechanisms. I think the experimental results are good enough to justify publication of this work in Nature Communications without the DFT calculations

Supplementary Information, 3: It is not given in the experimental description, how the corresponding products were further purified by column chromatography (e.g. used solvent, column material). It is only given, that "The hydrogen gas was released slowly and the solution was concentrated to give the crude products, which were purified by column chromatography to afford the final product." For the sake of reproducibility, there should be more information on the purification of the corresponding products provided to the reader.

Reviewer #2 (Remarks to the Author):

Although direct asymmetric reductive amination (ARA), providing a facile access to biologically active chiral amines, has achieved significant progress over the past decades, ARA using alkyl amines as the N-sources remains much less explored. In this manuscript, Chang and co-workers reported unexpected excellent results on the Ir-catalyzed ARA using primary alkyl amines as the N-sources. With the easily available and tunable Ir-monophos as catalysts, a wide range of substrates including aromatic and aliphatic ketones were well compatible, and most importantly, excellent enantioselectivities under low catalyst loading were achieved. The new method was also applied in the 10-gram-scale experiment and synthesis of several important pharmaceuticals. In addition, a reasonable catalytic mechanism was proposed based on DFT calculation. This finding is of great importance and represents a breakthrough in the asymmetric synthesis of chiral N-alkyl amines, and these results are valuable to synthetic community and suitable for the publication in this journal after some minor revisions.

1. As mentioned in the mechanism study, the amine substrate serves as a ligand of the Ir-catalyst. Did the authors try to prepare the Ir-complex containing an amine ligand? In addition, a related reference should be cited (Chem. Eur. J. 2012, 11578).
2. Although the results reported herein are the best, ADA using benzyl amines was reported previously (Ref. 38-39). Therefore, it is suggested for the authors to delete the phrase of “for the first time” from the Abstract.
3. Some related references are suggested to be cited. For example, Chem. Rev. 2022, 122,269 (a recent review paper on AH of imines); Chem. Eur. J. 2011, 17, 1109 (AH of N-alkyl imines).
4. For Ref. 38, the title of the paper should be added.

Reviewer #3 (Remarks to the Author):

Chang and coworkers have developed an Ir-catalyzed asymmetric reductive amination with primary alkyl amines as the N-sources, a reaction which is very useful and straightforward for the synthesis of pharmaceuticals. The substrate scope, regarding the ketone part, are remarkable since both aryl ketone and dialkyl ketones are well tolerated. In addition, the amine sources are also broad as various primary amines including functionalized amines are accommodated. Although the catalytic system is not new and have been used by themselves and also other groups, the reaction offers a very practical and highly enantioselective method to access chiral secondary amines of medicinal interest, thus will attract broad readership from both academy and industry. The authors also found the addition of FeCl₃ as additive had obvious positive effect on the enantiocontrol, this is actually quite intriguing. Considering the importance of the reaction and the excellent results under optimized conditions, I recommend the acceptance of this work after the listed questions being solved.

(1) The authors argued that their work was the “first time primary alkyl amines effectively serving as the N-sources” and “There have no primary alkyl amines been utilized as the N-coupling partners in transition-metal-catalyzed DARAs” in the manuscript, which is somehow inaccurate. The use of Ir/phosphoramidite complex as catalyst for ARA of ketones and alkyl amine has previously being studied, such as in ref *Org. Process. Res. Dev.* 2018, 22, 1817. In addition, alkyl amines (allylic amines) had been used in Ru-catalyzed asymmetric hydrogen transfer amination (see: *Adv. Synth. Catal.* 2005, 347, 1917 – 1920), which also belongs to ARA category. These refs should be added.

(2) In Scheme 1, eq C, the drawing gives me an impression that the Ir complex is at valence two? Similar problems existed in figure 1 as well. This kind of drawings will mislead the readers that the hydrogen transfer process was happened on a Ir(II) or Ir(I) species.

(3) In scheme 2A, for intermediate VI and also TS4, the Ir species should be anionic, otherwise it's not reasonable. These gave the readers impressions that the Ir(IV) species was involved.

(4) Regarding the mechanism, the authors proposed an outer sphere model, however, other routes are also possible. Is it possible that, the Cl⁻ was not binding to the Ir center, instead a Ir(III)H₃ species was formed, which subsequently transfer a hydride to protonated iminium species?

For the SI part: The SI was overall well prepared, but I do have some questions which could be figured out to enhance the quality.

(1) It's necessary to add the optical rotation data for all enantioenriched compounds since it's helpful when people want to assign the absolute configuration of similar compounds.

(2) The HPLC spectra of racemic samples of L4-I6 are suggested to be added.

For reviewer 1:

- 1) The role of the stoichiometric additive [Ti(OiPr)₄] is not discussed at all or at least described, why it was added in all reactions. The authors should add some description, why this was added (most likely to force imine formation by water scavenging) and what happens without it.

Thanks for the suggestion. Titanium isopropoxide is known to be able to facilitate the formation of the imine intermediates. We have added the discussion (P4, line 1) and corresponding reference (ref. 59).

- 2) On the role of the FeCl₃/HCl co-additives. The authors assume, that it's due to the bronsted acid behavior of the HCl or generated HCl from FeCl₃ which leads to the improved ee. But why is the ee not really increased, when adding the bronsted acid TsOH (entry 11 in table 1)? One could also assume, that there can be reactions with the [Ti(OiPr)₄] additive, as this is prone to hydrolysis. As the beneficial effect to improve the ee was only shown with the Cl-containing additives, may it be also the role of the Cl as a crucial ligand on the Ir to improve the ee? One can speculate, that by adding a Cl-source, the Ir-Cl species stay predominant, as the authors show in scheme 2, the Cl-ligand is crucial in the mechanism. It is also known from other ARA reports, that the halide ligand on the metal is of importance (see ref31). To validate this (or to rule this out), an experiment just adding a Cl-source without acidic properties as LiCl or CsCl would be of interest to see, if only adding Cl improves the ee, which could also be explained by the proposed mechanism. Or add HBr.

Thanks for the suggestion. We have carried the control experiments for the reaction of **1** and **2** (Table 1, entry 15). With the addition of LiCl or CsCl instead of FeCl₃, the results were nearly the same as those for entry 9, which indicates the added Cl⁻ does not have significant influence on the reaction. With the addition of 10 mol% Et₃NHBr, the results were almost the same as from FeCl₃. The above results indicate FeCl₃ mainly functions to release HCl. The Cl atom on the Ir-Cl complex in Scheme 2 (now Fig. 3) is from the metal precursor [Ir(cod)Cl]₂. As for the reason that TsOH did not work as well as FeCl₃, we think it results from the difference of their acidity.

- 3) DFT calculations/Mechanistic studies: In the calculations (see supplementary information 4), it is described, that "Single point energy calculations were performed on optimized geometries in EthylEthanoate solvent using the IEFPCM model at the B3LYP-D3 level of theory." This is a very unpolar solvent considered for the calculations. Especially, as significant ee where only obtained when using polar and protic solvents as MeOH or TFE. TFE is a rather polar and protic solvent and as crucial intermediated in the proposed mechanism are charged and H-interactions are

involved, one can expect, that the energies would also be affected, if a more polar-protic solvent environment is considered in the DFT calculations (e.g. the proposed ion pairs should be less stable in a very polar-protic environment than in a unipolar solvent). The authors should at least discuss, why they assume, that this rather unipolar solvent environment in the DFT calculations will provide accurate energies for the proposed mechanism. But I would recommend recalculating using a more realistic solvent model.

Thanks for the suggestion. We have carried out the calculation for the reaction in $\text{CF}_3\text{CH}_2\text{OH}$ and found the activation energies are truly lower than those in EtOAc. We have updated the mechanism scheme with the new results (now Fig. 3) and put the comparison (S.Fig. 66) and related data to the Supplementary Information.

- 4) In Scheme 2 and the according discussion, it is nowhere provided, what is the solvent coordinated in the species I, II and III. This should be provided. Species IV is a neutral Ir(III) dihydride. But after the H₂ activation, species VI and the following transition state it should be an anionic Ir(III)-trihydrides. All relevant charges should be provided in the scheme.

Thanks for the suggestion. The solvent is $\text{CF}_3\text{CH}_2\text{OH}$. We have added the solvent to the mechanism and changed the charge for complex **VI**, transition states **TS4-R** and **TS4-S** to -1 in Fig. 3.

- 5) There is a general inconsistency between the structural tuning in figure 1 and the proposed calculated mechanism in scheme 2: In figure 1, the structural tuning is described on square-planar tetrahedral Ir(I) (?) complexes, but in the proposed mechanism, all relevant Ir-species responsible for the enantioselectivity are octahedral Ir(III) complexes. In this context, was any pathway calculated or considered, where a H₂ activation via a Ir(I)-Ir(III) mechanism is involved (including a reductive elimination step)? Also, the pictures of provided calculated transition states TS4 should be improved, to provide the reader valuable information. In TS4-S the MeNH₂ ligand looks just like a NH₃. Where is the Methyl-group? To add some drawing (dashed lines) for the relevant bonding situation incl. bond length in TS4-R would be a useful information. A picture of the structure with the relevant bond length in VI would also be of interest. Is the positive charge still on the H, as drawn in the schematic picture? Seems rather unusual for a structure which is not a transition state. Therefore, I would recommend, to revise the calculations according to the above given points and also provide (e.g. in the supporting information) further

insights and alternative pathways considered. But this would result in a lot of additional work, may better address in a follow up paper focusing on the mechanisms. I think the experimental results are good enough to justify publication of this work in Nature Communications without the DFT calculations.

Thanks for the suggestion. We have carried out the calculations for an Ir(I)-Ir(III) catalytic cycle and found the major activation energy is higher than the proposed “outer-sphere” H-addition catalytic cycles, in which Ir remains at +3. We have added the results (S.Fig. 67) and related data to the Supplementary Information. We have corrected the square-planar Ir complexes to octahedral Ir(III) complexes in Figure 1 (now Fig. 2). We have added some key bond length to Scheme 2 (now Fig. 3).

- 6) Supplementary Information, 3: It is not given in the experimental description, how the corresponding products were further purified by column chromatography (e.g. used solvent, column material). It is only given, that “The hydrogen gas was released slowly and the solution was concentrated to give the crude products, which were purified by column chromatography to afford the final product.”. For the sake of reproducibility, there should be more information on the purification of the corresponding products provided to the reader.

Thanks for the suggestion. We have added the detailed purification conditions to the Methods section and the Supplementary Information.

For reviewer 2:

- 1) As mentioned in the mechanism study, the amine substrate serves as a ligand of the Ir-catalyst. Did the authors try to prepare the Ir-complex containing an amine ligand? In addition, a related reference should be cited (Chem. Eur. J. 2012, 11578).

Thanks for the suggestion. In the reaction of acetophenone **1** and 3-phenylpropylamine **2**, we tried a few control experiments by mixing the Ir-**L4** complex with several amines, including methylamine, aniline and benzylamine, and found for the results there was no significant difference between them and the standard conditions. We thought it is probable the chiral phosphoramidite ligand rather than the methylamine ligand which controls the enantioselectivity. We have added the above reference to the manuscript (ref. 69).

- 2) Although the results reported herein are the best, ADA using benzyl amines was reported previously (Ref. 38-39). Therefore, it is suggested for the authors to delete the phrase of “for the first time” from the Abstract.

Thanks for the suggestion. We have deleted “for the first time” and added two references to the manuscript (ref. 45&46).

- 3) Some related references are suggested to be cited. For example, Chem. Rev. 2022, 122,269 (a recent review paper on AH of imines); Chem. Eur. J. 2011, 17, 1109 (AH of N-alkyl imines).

Thanks for the suggestion. We have added the above two references (ref. 9&50).

- 4) For Ref. 38, the title of the paper should be added.

Thanks for the suggestion. We have added the title to ref. 38.

For reviewer 3:

- 1) The authors argued that their work was the “first time primary alkyl amines effectively serving as the N-sources” and “There have no primary alkyl amines been utilized as the N-coupling partners in transition-metal-catalyzed DARAs” in the manuscript, which is somehow inaccurate. The use of Ir/phosphoramidite complex as catalyst for ARA of ketones and alkyl amine has previously being studied, such as in ref Org. Process. Res. Dev 2018. 22. 1817. In addition, alkyl amines (allylic amines) had been used in Ru-catalyzed asymmetric hydrogen transfer amination (see: Adv. Synth. Catal. 2005, 347, 1917 – 1920), which also belongs to ARA category. These refs should be added.

Thanks for the suggestion. We have rewritten the corresponding parts and added two new references (ref. 45&46).

- 2) In Scheme 1, eq C, the drawing gives me an impression that the Ir complex is at valence two? Similar problems existed in figure 1 as well. This kind of drawings will mislead the readers that the hydrogen transfer process was happened on a Ir(II) or Ir(I) species.

Thanks for the suggestion. For both Ir complexes the oxidation state are at +3. We have modified the figures in Scheme 1 (now Fig. 2) from square-planar to octahedral, the structure of complex **III** and added one negative charge to complex **VI**, transition states **TS4-R** and **TS4-S** of Scheme 2 (now Fig. 3).

- 3) In scheme 2A, for intermediate VI and also TS4, the Ir species should be anionic, otherwise it's not reasonable. These gave the readers impressions that the Ir(IV) species was involved.

Thanks for the suggestion. We have added one negative charge to complex **VI**, transition states **TS4-R** and **TS4-S** of Scheme 2 (now Fig. 3).

- 4) Regarding the mechanism, the authors proposed an outer sphere model, however, other routes are also possible. Is it possible that, the Cl⁻ was not binding to the Ir center, instead a Ir(III)H₃ species was formed, which subsequently transfer a hydride to protonated iminium species?

Thanks for the suggestion. We have carried out the calculations for an Ir(I)-Ir(III) catalytic cycle and found the major activation energy is higher than the proposed “outer-sphere” H-addition, in which Ir remains at +1. We have added the results (S.Fig. 67) and related data to the Supplementary Information. Cl⁻ on the Ir-Cl complex in Scheme 2 (now Fig. 3) is from the metal precursor [Ir(cod)Cl]₂. From calculation, the Gibbs free energy difference between Ir(III)H₂Cl and Ir(III)H₃ is -14.4 kcal/mol. That is, to replace Cl with H is endergonic. Thus, we think it is difficult energetically for this transformation to occur.

- 5) For the SI part: The SI was overall well prepared, but I do have some questions which could be figured out to enhance the quality.
- (1) It's necessary to add the optical rotation data for all enantioenriched compounds since it's helpful when people want to assign the absolute configuration of similar compounds.
 - (2) The HPLC spectra of racemic samples of L4-L6 are suggested to be added.

Thanks for the suggestion. We have added the optical rotation data for all enantioenriched compounds. For the chiral ligands **L4-L6**, we tried all 5 available chiral HPLC columns but could not found the separation method.

REVIEWERS' COMMENTS

Reviewer #1 (Remarks to the Author):

This is a revised manuscript, where I was reviewer 1 on the initial submission. Therefore, I only respond to the authors responds to my comments. But as far as I can see, also the comments of the other two reviewer were addressed well in this revision. Generally, I thank the authors for the additional work to address all these comments. It further improved this manuscript and made it clearer at certain points. Therefore, I think it is now suitable to be accepted for publication in Nature Communications. Nice work!

For reviewer 1:

1) The role of the stoichiometric additive [Ti(OiPr)₄] is not discussed at all or at least described, why it was added in all reactions. The authors should add some description, why this was added (most likely to force imine formation by water scavenging) and what happens without it.

Thanks for the suggestion. Titanium isopropoxide is known to be able to facilitate the formation of the imine intermediates. We have added the discussion (P4, line 1) and corresponding reference (ref. 59).

Respond by reviewer 1: Fine. Thanks for the addition of this sentence and reference.

2) On the role of the FeCl₃/HCl co-additives. The authors assume, that it's due to the bronsted acid behavior of the HCl or generated HCl from FeCl₃ which leads to the improved ee. But why is the ee not really increased, when adding the bronsted acid TsOH (entry 11 in table 1)? One could also assume, that there can be reactions with the [Ti(OiPr)₄] additive, as this is prone to hydrolysis. As the beneficial effect to improve the ee was only shown with the Cl-containing additives, may it be also the role of the Cl as a crucial ligand on the Ir to improve the ee? One can speculate, that by adding a Cl-source, the Ir-Cl species stay predominant, as the authors show in scheme 2, the Cl-ligand is crucial in the mechanism. It is also known from other ARA reports, that the halide ligand on the metal is of importance (see ref31). To valide this (or to rule this out), an experiment just adding a Cl-source without acidic properties as LiCl or CsCl would be of interest to see, if only adding Cl improves the ee, which could also be explained by the proposed mechanism. Or add HBr.

Thanks for the suggestion. We have carried the control experiments for the reaction of 1 and 2 (Table 1, entry 15). With the addition of LiCl or CsCl instead of FeCl₃, the results were nearly the same as those for entry 9, which indicates the added Cl⁻ does not have significant influence on the reaction. With the addition of 10 mol% of Et₃N.HBr, the results were almost the same as from FeCl₃. The above results indicate FeCl₃ mainly functions to release HCl. The Cl atom on the Ir-Cl complex in Scheme 2 (now Fig. 3) is from the metal precursor [Ir(cod)Cl]₂. As for the reason that TsOH did not work as well as FeCl₃, we think it results from the difference of their acidity.

Respond by reviewer 1: Thanks for adding this experiment. This answers my question on this point.

3) DFT calculations/Mechanistic studies: In the calculations (see supplementary information 4), it is described, that "Single point energy calculations were performed on optimized geometries in EthylEthanoate solvent using the IEFPCM model at the B3LYP-D3 level of theory." This is a very unpolar solvent considered for the calculations. Especially, as significant ee were only obtained when using polar and protic solvents as MeOH or TFE. TFE is a rather polar and protic solvent and as crucial intermediates in the proposed mechanism are charged and H-interactions are involved, one can expect, that the energies would also be affected, if a more polar-protic solvent environment is considered in the DFT calculations (e.g. the proposed ion pairs should be less stable in a very polar-protic environment than in a unipolar solvent). The authors should at least discuss, why they assume, that this rather unipolar solvent environment in the DFT calculations will provide accurate energies for the proposed mechanism. But I would recommend recalculating using a more realistic solvent model.

Thanks for the suggestion. We have carried out the calculation for the reaction in CF₃CH₂OH and found the activation energies are truly lower than those in EtOAc. We have updated the mechanism scheme with the new results (now Fig. 3) and put the comparison (S.Fig. 66) and related data to the Supplementary Information.

Respond by reviewer 1: Thanks for recalculating the energies with CF₃CH₂OH as the solvent. This energies should now be more accurate than the ones before in the less polar solvent.

4) In Scheme 2 and the according discussion, it is nowhere provided, what is the solvent coordinated in the species I, II and III. This should be provided. Species IV is a neutral Ir(III) dihydride. But after the H₂ activation, species VI and the following transition state it should be an anionic Ir(III)-trihydrides. All relevant charges should be provided in the scheme.

Thanks for the suggestion. The solvent is CF₃CH₂OH. We have added the solvent to the mechanism and changed the charge for complex VI, transition states TS4-R and TS4-S to -1 in Fig. 3.

Respond by reviewer 1: Fine.

5) There is a general inconsistency between the structural tuning in figure 1 and the proposed calculated mechanism in scheme 2: In figure 1, the structural tuning is described on square-planar tetrahedral Ir(I) (?) complexes, but in the proposed mechanism, all relevant Ir-species responsible for the enantioselectivity are octahedral Ir(III) complexes. In this context, was any pathway calculated or considered, where a H₂ activation via a Ir(I)-Ir(III) mechanism is involved (including a reductive elimination step)? Also, the pictures of provided calculated transition states TS4 should be improved, to provide the reader valuable information. In TS4-S the MeNH₂ ligand looks just like a NH₃. Where is the Methyl-group? To add some drawing (dashed lines) for the relevant bonding situation incl. bond length in TS4-R would be a useful information. A picture of the structure with the relevant bond length in VI would also be of interest. Is the positive charge still on the H, as drawn in the schematic picture? Seems rather unusual for a structure which is not a transition state. Therefore, I would recommend, to revise the calculations according to the above given points and also provide (e.g. in the supporting information) further insights and alternative pathways considered. But this would result in a lot of additional work, may better address in a follow up paper focusing on the mechanisms. I think the experimental results are good enough to justify publication of this work in Nature Communications without the DFT calculations.

Thanks for the suggestion. We have carried out the calculations for an Ir(I)-Ir(III) catalytic cycle and found the major activation energy is higher than the proposed "outer-sphere" H-addition catalytic cycles, in which Ir remains at +3. We have added the results (S.Fig. 67) and related data to the Supplementary Information. We have corrected the square-planar Ir complexes to octahedral Ir(III) complexes in Figure 1 (now Fig. 2). We have added some key bond length to Scheme 2 (now Fig. 3).

Respond by reviewer 1: Thanks for the addition of this, it makes it way clearer for the reader.

6) Supplementary Information, 3: It is not given in the experimental description, how the corresponding products were further purified by column chromatography (e.g. used solvent, column material). It is only given, that "The hydrogen gas was released slowly and the solution was concentrated to give the crude products, which were purified by column chromatography to afford the final product." For the sake of reproducibility, there should be more information on the purification of the corresponding products provided to the reader.

Thanks for the suggestion. We have added the detailed purification conditions to the Methods section and the Supplementary Information.

Respond by reviewer 1: Fine. Thanks for the addition of this information.

Reviewer #2 (Remarks to the Author):

According to the reviewers' comments, sufficient changes and additions were made by the authors, which increase the overall quality of the new version of the manuscript. I thus recommend its publication in this Journal.

Reviewer #3 (Remarks to the Author):

I am satisfied with the revision. However, there still exist several typos which could be corrected. Such as, in fig 3, SH should be replaced with "S" in structure II.; for entry c in fig 1, it is more accurate to draw the Ir complex as octahedral configuration, instead of square-planar to octahedral.

For reviewer 3:

- 1) I am satisfied with the revision. However, there still exist several typos which could be corrected. Such as, in fig 3, SH should be replaced with "S" in structure II.; for entry c in fig 1, it is more accurate to draw the Ir complex as octahedral configuration, instead of square-planar to octahedral.

Thanks for the suggestion. We have changed "SH" in Fig. 3 (now Fig 5) to "S", the square-planar configuration of the Ir complexes to octahedral in Fig 1.